# Serum Levels of Stromal Cell-Derived Factor-1α and Vascular Endothelial Growth Factor Predict Clinical Outcomes in Head and Neck Squamous Cell Carcinoma Patients Receiving TPF Induction Chemotherapy

**DOI:** 10.3390/biomedicines10040803

**Published:** 2022-03-29

**Authors:** Yen-Hao Chen, Chih-Yen Chien, Yu-Ming Wang, Shau-Hsuan Li

**Affiliations:** 1Division of Hematology-Oncology, Department of Internal Medicine, Kaohsiung Chang Gung Memorial Hospital, College of Medicine, Chang Gung University, Kaohsiung 833, Taiwan; alex2999@cgmh.org.tw or; 2School of Medicine, College of Medicine, Chang Gung University, Taoyuan 333, Taiwan; 3School of Medicine, Chung Shan Medical University, Taichung 402, Taiwan; 4Department of Otolaryngology, Kaohsiung Chang Gung Memorial Hospital, College of Medicine, Chang Gung University, Kaohsiung 833, Taiwan; cychien3965@cgmh.org.tw; 5Department of Radiation Oncology, Kaohsiung Chang Gung Memorial Hospital, College of Medicine, Chang Gung University, Kaohsiung 833, Taiwan; scorpion@cgmh.org.tw

**Keywords:** SDF-1α, VEGF, head and neck cancer, squamous cell carcinoma, induction chemotherapy, TPF

## Abstract

Chemokines, such as stromal cell-derived factor-1α (SDF-1α) and vascular endothelial growth factor (VEGF), are associated with clinical outcomes in several cancer types. This study aimed to investigate the role of SDF-1α and VEGF in the prognosis of patients with head and neck squamous cell carcinoma (HNSCC) who underwent TPF induction chemotherapy (docetaxel, cisplatin, and 5-fluorouracil). A total of 77 HNSCC patients were enrolled and circulating SDF-1α and VEGF values were examined at two time points for each patient, including pre-TPF treatment (treatment-naïve) and post-TPF treatment but before chemoradiotherapy. The median progression-free survival (PFS) and overall survival (OS) were 18.1 and 32.9 months, respectively. Decreased SDF-1α and VEGF levels after TPF treatment, post-TPF SDF-1α < 1500 pg/mL and VEGF value < 150 pg/mL were independent prognostic factors for better PFS and OS in univariate and multivariate analyses. A combination of SDF-1α and VEGF values may predict clinical outcomes significantly. Our study confirmed the role of SDF-1α and VEGF in the disease progression of HNSCC, and that decreased SDF-1α and VEGF after TPF treatment and lower post-TPF SDF-1α and VEGF values were associated with better prognosis in HNSCC patients who received induction chemotherapy with TPF followed by chemoradiotherapy.

## 1. Introduction

Head and neck squamous cell carcinoma (HNSCC) is one of the most aggressive malignancies worldwide and is the sixth leading cause of mortality in Taiwan [1]. Most patients with HNSCC have a locally advanced status at diagnosis, and multidisciplinary therapeutic modalities, including chemotherapy, radiotherapy and surgical resection, may be necessary. Growing evidence has confirmed the crucial role of chemotherapy combined with radiotherapy in locally advanced HNSCC, especially in patients whose tumors are unresectable or require organ preservation [2,3,4]. As chemoradiotherapy (CRT) improves local control of HNSCC, distant metastasis has become an important issue and is considered a critical cause of treatment failure. Systemic chemotherapy, also known as induction chemotherapy, has been developed to reduce the incidence of distant metastases and improve local tumor control. Recently, two large randomized controlled trials have shown that induction chemotherapy with docetaxel, cisplatin, and 5-fluorouracil (TPF) provides superior progression-free survival (PFS) and overall survival (OS) compared to induction chemotherapy with cisplatin and 5-fluorouracil (PF), suggesting that TPF is the most effective regimen for induction chemotherapy [5,6,7].

Chemokines consist of many small, secreted proteins which are involved in tumor cell proliferation, invasion, migration and metastasis, contributing to disease progression. Stromal cell-derived factor-1α (SDF-1α) is a homeostatic chemokine and expressed in multiple organs; C-X-C chemokine receptor type 4 (CXCR4) is a receptor of SDF-1α, and the SDF-1α/CXCR4 pathway activation modulates the downstream signaling pathway, leading to chemotaxis, anti-apoptosis, tumor cell proliferation, and gene transcription [8,9]. Growing evidence has shown that the SDF-1α/CXCR4 axis plays an important role in several cancer types. Albert et al. reported that the SDF-1α/CXCR4 axis is related to disease progression, high metastatic potential and poor outcomes in HNSCC [10]. In lung cancer study, overexpression of SDF-1α is associated with distant metastasis and promotes disease progression through modulating tumor stem cells [11]. Amara et al. showed that high expression of the couple SDF-1/CXCR4 enhances liver metastasis, and correlated with poor prognosis in colorectal cancer patients [12]. Our previous study also confirmed that overexpression of SDF-1α was significantly associated with worse disease-free survival (DFS) and OS [13].

Angiogenesis is a complex process that involves cellular and molecular interactions between cancer cells and their microenvironment, which consists of growth factors, cytokines, chemokines and the extracellular matrix. Angiogenesis also plays a critical role in cancer progression, including tumor growth, maintenance and metastasis. Some historical observations have revealed that tumor growth depends on angiogenesis through the proliferation of new blood vessel networks that supply a tumor and its microenvironment with oxygen and nutrients, contributing to tumor growth, invasion, migration and distant metastasis [14]. Vascular endothelial growth factor (VEGF) is an important angiogenic factor, and VEGF signaling is crucial to angiogenesis and tumor progression, such as the proliferation of endothelial cells, vascular hyperpermeability, initiation of carcinogenesis, and immune modulation [15,16]. A lot of studies have shown that elevated baseline VEGF levels are associated with poor prognosis in many cancer types, including small cell lung cancer, non-small cell lung cancer, colorectal cancer and renal cell carcinoma [17,18,19,20]. Moreover, our previous study also demonstrated that kinetic changes to lower post-treatment VEGF levels and decreased VEGF after treatment represent prognostic factors for superior clinical outcomes in esophageal cancer patients who underwent CRT [21]. Moreover, Wang et al. reported that serum VEGF is a biomarker that correlates with neoadjuvant chemotherapy response in triple-negative breast cancer, including the predictive value of pathological complete response to neoadjuvant chemotherapy and DFS [22].

In general, induction chemotherapy with TPF, the powerful regimen, incorporated both induction chemotherapy with PF and CRT in an attempt to improve locoregional control, eliminate distant metastases, and prolong survival. However, the role of SDF-1α and VEGF in HNSCC induction chemotherapy remains unclear. The aim of the present study was to explore the role of SDF-1α and VEGF in the prognosis of patients with HNSCC who underwent induction chemotherapy with TPF.

## 2. Materials and Methods

### 2.1. Patient Population

We retrospectively reviewed the medical records of patients with HNSCC who received TPF induction chemotherapy at Kaohsiung Chang Gung Memorial Hospital between January 2010 and December 2016. First, we excluded patients who had a history of a second primary malignancy. Second, patients who had distant metastasis, and who underwent any anti-cancer treatment before induction chemotherapy or received induction chemotherapy regimens other than TPF were also excluded. Patients enrolled in our study had an Eastern Cooperative Oncology Group Scale of Performance Status (ECOG PS) of 0 or 1. Finally, 77 patients with HNSCC were identified. 

### 2.2. Induction Chemotherapy with TPF

The TPF chemotherapy consisted of docetaxel (60 mg/m^2^; 1-h intravenous infusion), followed by intravenous cisplatin (60 mg/m^2^; 3-h infusion), and 5-fluorouracil (600 mg/m^2^; 24-h infusion) for four days every three weeks. Induction chemotherapy with TPF was administered for three cycles, except in cases of disease progression, intolerance to chemotherapy toxicity, or a withdrawal of consent by patients. Chemotherapy was administered according to a previously described protocol [23,24].

### 2.3. Chemoradiotherapy Planning 

All patients received CRT four weeks after the start of the third cycle of TPF induction chemotherapy. The details of radiotherapy were as follows: each patient underwent CT simulation with images from the upper neck through the upper abdomen with a 5 mm slice thickness, and patients were immobilized in customized thermoplastic devices. The three-dimensional conformal radiotherapy technique or the intensity-modulated radiotherapy technique with 6-MV or 10-MV photon beams were used for treatment planning and radiation delivery. The curative radiotherapy dose to the primary tumor was administered at 70 Gy in 35 daily fractions of 2 Gy 5 days per week. The doses administered to the involved lymph nodes were between 60–70 Gy, and at least 50 Gy to uninvolved lymph nodes. Cisplatin (40 mg/m^2^; 2-h infusion) was administered intravenously weekly at a maximum of seven doses during the course of radiotherapy.

### 2.4. Serum SDF-1α and VEGF Measurement

Circulating levels of SDF-1α and VEGF in peripheral blood samples were examined using a commercially available enzyme-linked immunosorbent assay (ELISA) kit (Quantikine; R&D Systems, Abingdon, UK) for each patient in our study. Peripheral venous blood samples were drawn and collected in sterile test tubes, and the serum was separated, aliquoted, and stored at −20 °C until use. Samples were measured in duplicate, and the mean value was determined as the final concentration according to the manufacturer’s instructions. The ELISA plates were analyzed using an Emax precision microplate reader (Molecular Devices, Sunnyvale, CA, USA). Subsequently, standard curves were generated, and serum SDF-1α and VEGF values were determined. Validation of the intra- and inter-assays was also performed. A total of 2 SDF-1α and VEGF levels were examined for each patient, including point 1: pre-TPF treatment (treatment-naïve, TPF induction chemotherapy cycle 1 day 1, one hour before chemotherapy) and point 2: post-TPF treatment but before CRT (CRT day 1, one hour before chemotherapy and radiotherapy). Zajac et al. reported that the mean SDF-1α level for esophageal squamous cell carcinoma patients was 1500 pg/mL so a value of 1500 pg/mL was considered as the cut-off level in our study [25]. The mean VEGF level for HNSCC patients was found to be 144.5 pg/mL in a previous study; therefore, a value of 150 pg/mL was regarded as the cut-off level in our study [26]. 

### 2.5. Ethics Statement

This retrospective study was approved by the Chang Gung Medical Foundation institutional review board (103-3342B and 201900561B0). All procedures used in studies involving human participants were performed in accordance with the ethical standards of the Institutional Research Committee and the World Medical Association Declaration of Helsinki. Written informed consent was obtained from each patient, and all methods were performed in accordance with approved guidelines.

### 2.6. Statistical Analysis

All statistical analyses were performed using SPSS software v.22 (International Business Machines Corp., Armonk, NY, USA). Survival analysis was conducted using the Kaplan–Meier method, and differences were tested using the log-rank test. A Cox proportional hazards model using the stepwise method was used to estimate the independent prognostic factors in multivariate analysis. Hazard ratios (HRs) with 95% confidence intervals (CIs) and *p*-values were calculated to assess the strength of the association between survival and prognostic parameters. The PFS was calculated from the start of TPF induction chemotherapy until tumor recurrence or death from any cause, without evidence of recurrence. The OS was defined as the time from HNSCC diagnosis to death or to the time of last living contact. All tests were two-sided, and statistical significance was set at *p* < 0.05.

## 3. Results

### 3.1. Patient Characteristics

Between January 2010 and December 2016, 77 patients with HNSCC received TPF induction chemotherapy, followed by CRT at Kaohsiung Chang Gung Memorial Hospital. The ECOG PS was ≤1 for all patients. Among the 77 patients, there were 73 male patients and 4 were female, with a median age of 53 years (range: 29–82 years). The distribution of the primary tumor location was: oral cavity in 22 patients (28.6%), oropharynx in 28 patients (36.4%), hypopharynx in 11 patients (14.3%), and larynx in 16 patients (20.7%). Human papillomavirus (HPV) status was positive in 6 patients (7.8%) and negative in 71 patients (92.2%). Nineteen patients (24.7%) had clinical T2 status, 9 patients (11.7%) had T3 status, 49 patients (63.6%) had T4 status, 18 patients (23.4%) did not have lymph node metastasis, 12 patients (15.6%) had clinical N1 status, 41 patients (24.7%) had clinical N2 status, and 6 patients (7.8%) had clinical N3 status. Tumor stage data showed that five patients (6.5%) had stage II, seven patients (9.1%) had stage III, 34 patients (44.2%) had stage IVA, and 31 patients (40.2%) had stage IVB. The distribution of histological grade included grade 1 in 19 patients (24.7%), grade 2 in 43 patients (55.8%) and grade 3 in 15 patients (19.5%). The characteristics of these 77 HNSCC patients are shown in Table 1.

In our study, the median follow-up period was 114.2 months for 13 survivors and 32.9 months for all 77 patients. The mean and standard deviation of SDF-1α values of pre-TPF treatment and post-TPF treatment were 1885.5 ± 504.2 pg/mL and 1625.6 ± 325.9 pg/mL, respectively. On the other hand, the mean and standard deviation of VEGF values of pre-TPF treatment and post-TPF treatment showed 189.6 ± 139.2 pg/mL and 155.0 ± 106.4 pg/mL, respectively (Figure 1).

### 3.2. SDF-1α, VEGF and Clinical Outcomes

In our study, the median PFS and OS were 18.1 months and 32.9 months, respectively. 

In the PFS analysis, there were no statistically significant differences in parameters, such as age, gender, tumor location, clinical lymph node status, and clinical tumor stage. The 58 patients who had clinical T2 were noted to have better PFS than the 19 patients who had clinical T3–4 disease (71.3 months versus 13.9 months, *p* = 0.003). Better PFS was mentioned in 6 patients with positive HPV status in comparison with the other 71 patients with negative HPV status (116.6 months versus 16.1 months, *p* = 0.014). Significantly superior PFS was identified in 42 patients who exhibited decreased VEGF levels after TPF treatment compared to 35 patients without decreased VEGF levels (38.7 months versus 9.9 months, *p* = 0.001, Figure 2A). The 21 patients with decreased SDF-1α levels after TPF treatment had better PFS than the other 56 patients without decreased SDF-1α levels (116.6 months versus 11.2 months, *p <* 0.001, Figure 2B). The 36 patients with post-TPF VEGF abundance < 150 pg/mL had longer PFS compared to the other 41 patients with post-TPF VEGF ≥ 150 pg/mL (41.0 months versus 13.3 months, *p* = 0.002, Figure 2C). Better OS was found in 18 patients who exhibited post-TPF SDF-1α abundance < 1500 pg/mL in comparison with the other 59 patients who had post-TPF SDF-1α ≥ 1500 pg/mL (115.1 months versus 14.0 months, *p <* 0.001, Figure 2D). In multivariate analysis, a VEGF decrease after TPF treatment (HR: 0.46, 95% CI: 0.27–0.52, *p* = 0.003), SDF-1α decrease after TPF treatment (HR: 0.38, 95% CI: 0.18–0.77, *p* = 0.007), post-TPF VEGF < 150 pg/mL (HR: 0.50, 95% CI: 0.29–0.86, *p* = 0.011) and post-TPF SDF-1α < 1500 pg/mL (HR: 0.43, 95% CI: 0.19–0.95, *p* = 0.036) were independent prognostic parameters for better PFS.

With respect to OS, the univariate analysis revealed that age, gender, clinical lymph node status, and clinical tumor stage were not statistically significant predictors of OS. Fifty-five patients with primary tumors located in the oropharynx, hypopharynx and larynx had better OS than the remaining 22 patients with tumors in the oral cavity (35.7 months versus 16.3 months, *p* = 0.032). Superior OS was found in the 58 patients with clinical T2 disease compared to the 19 patients with clinical T3–4 disease (71.3 months versus 25.6 months, *p* = 0.003). Better OS was mentioned in 6 patients with positive HPV status than the rest of the 71 patients with negative HPV status (not reached versus 28.6 months, *p* = 0.007). Forty-two patients who exhibited decreased VEGF levels after TPF treatment had longer OS in comparison with 35 who were without decreased VEGF levels (55.4 months versus 18.4 months, *p* = 0.002, Figure 3A). The 21 patients with decreased SDF-1α levels after TPF treatment were mentioned to have superior OS than the other 56 patients without decreased SDF-1α levels (116.6 months versus 23.3 months, *p <* 0.001, Figure 3B). Significantly better OS was noted in 36 patients with post-TPF VEGF levels < 150 pg/mL than in the other 41 patients with post-TPF VEGF ≥ 150 pg/mL (58.3 months versus 20.3 months, *p <* 0.002, Figure 3C). The 18 patients who exhibited post-TPF SDF-1α abundance < 1500 pg/mL had longer OS compared to the other 59 patients who had post-TPF SDF-1α ≥ 1500 pg/mL (not reached versus 27.0 months, *p <* 0.001, Figure 3D). Multivariate analysis showed that an age < 60 years (HR: 0.47, 95% CI: 0.24–0.90, *p* = 0.024), VEGF decrease after TPF treatment (HR: 0.43, 95% CI: 0.25–0.74, *p* = 0.002), SDF-1α decrease after TPF treatment (HR: 0.40, 95% CI: 0.20–0.83, *p* = 0.013), post-TPF VEGF < 150 pg/mL (HR: 0.38, 95% CI: 0.22–0.65, *p* = 0.001) and post-TPF SDF-1α < 1500 pg/mL (HR: 0.42, 95% CI: 0.18–0.95, *p* = 0.037) were independent prognostic factors for better OS. The survival outcomes of the univariate and multivariate analyses are shown in Table 2 and Table 3.

### 3.3. Combination of SDF-1α and VEGF and Prognosis

In the analysis of the kinetic change of VEGF and SDF-1α, the 77 HNSCC patients were divided into three groups, including group 1: patients without both VEGF and SDF-1α decreased after TPF, group 2: did not meet the criteria of group 1 and group 3, and group 3: patients with both VEGF and SDF-1α decreased after TPF. The PFS and OS among these three groups were significantly different. The PFS were 8.7 months, 26.8 months and 116.6 months in group 1, 2 and 3, respectively (group 1 versus group 2: *p* = 0.001, group 1 versus group 3: *p <* 0.001, group 2 versus group 3: *p* = 0.010, Figure 4A). Moreover, the OS showed 13.1 months in group 1, 37.2 months in group 2 and 116.6 months in group 3 (group 1 versus group 2: *p* = 0.002, group 1 versus group 3: *p <* 0.001, group 2 versus group 3: *p* = 0.012, Figure 4B).

With respect to the post-TPF SDF-1α and VEGF values, the 77 HNSCC patients were divided into three groups, including group 1: patients with both post-TPF VEGF ≥ 150 pg/mL and SDF-1α ≥ 1500 pg/mL, group 2: did not meet the criteria of group 1 and group 3, and group 3: patients without both post-TPF VEGF ≥ 150 pg/mL and SDF-1α ≥ 1500 pg/mL. There was a significant difference in PFS and OS among these three groups. The PFS showed 12.5 months in group 1, 16.1 months in group 2 and 115.1 months in group 3 (group 1 versus group 2: *p* = 0.026, group 1 versus group 3: *p <* 0.001, group 2 versus group 3: *p* = 0.003, Figure 4C). On the other hand, the OS were 18.8 months, 35.7 months and not reached in group 1, 2 and 3, respectively (group 1 versus group 2: *p* = 0.009, group 1 versus group 3: *p <* 0.001, group 2 versus group 3: *p* = 0.001, Figure 4D).

## 4. Discussion

Growing evidence has confirmed that induction chemotherapy with TPF provides a survival benefit in terms of patient PFS and OS compared to PF in HNSCC cases [5,6,7]. In addition, induction chemotherapy improves local control and reduces the incidence of distant metastases. Angiogenesis plays a crucial role in cancer progression, including carcinogenesis, proliferation, maintenance and metastasis. VEGF is a critical angiogenic factor involved in complex interactions involving cancer cells and their microenvironment. Several studies have shown that VEGF levels are associated with poor prognosis in many cancer types [17,18,19,20]. On the other hand, SDF-1α is a kind of homeostatic chemokine and is associated with many cancer types. Several studies have demonstrated that the activation of the SDF-1α/CXCR4 axis enhances tumor growth, invasion and gene transcription; and the inhibition of the SDF-1α/CXCR4 signal pathway reverses this phenomenon, contributing to apoptosis, cell cycle arrest, and malignant properties [10,27,28,29]. In the current study, patients with decreased SDF-1α and VEGF levels after TPF treatment and lower post-TPF SDF-1α and VEGF values were noted to have superior PFS and OS compared to those who did not, suggesting that SDF-1α and VEGF are both independent prognostic factors in patients with HNSCC who underwent induction chemotherapy with TPF.

The dose of TPF induction chemotherapy is an important factor. The clearance of docetaxel is associated with age, liver function, α1-acid glycoprotein levels, and body surface area. The originally recommended dose of docetaxel was 75 mg/m^2^ every three weeks in the TAX 324 trial [6]. However, the pharmacokinetics of docetaxel differ between Western and Asian populations and other ethnicities. For example, when docetaxel was administered to Japanese patients at a dose of 60 mg/m^2^, hematological toxicity events were more frequent and severe compared to a Western population treated with 75 mg/m^2^ [30]. In addition, another study also demonstrated that the incidence of docetaxel-induced severe neutropenia in Asian and non-Asian population clinical studies was significantly different, indicating that the role of ethnic diversity in docetaxel toxicity should be considered when interpreting the results of clinical trials [31]. Moreover, even when TPF induction chemotherapy was administered at a reduced dose, hematological toxicity events were still reported for patients with HNSCC [32,33]. In general, the Asian population is more susceptible to docetaxel toxicity, but the optimal dose of docetaxel remains unclear. Therefore, the TPF dose in our study was lower than that in the TAX 324 trial, and the current doses (docetaxel 60 mg/m^2^, cisplatin 60 mg/m^2^, and 5-fluorouracil 600 mg/m^2^) were determined based on previous studies [23,24]. 

Several studies have confirmed the relationship between angiogenesis and treatment response to chemotherapy in many types of malignancies. Serum VEGF levels decrease after chemotherapy, contributing to the enhancement of the curative effect in patients with advanced colorectal cancer [34]. Dirix et al. reported that serum VEGF values are higher in patients with disease progression than in those exhibiting a response to treatment in metastatic cancer patients, including breast, colorectal, ovarian, and renal carcinomas [35]. Hyodo et al. also demonstrated that patients with low circulating VEGF levels have a higher treatment response rate to chemotherapy than those with high VEGF levels in gastric and colorectal cancer [36]. In addition, VEGF was found to be correlated with the treatment response to neoadjuvant chemotherapy and is regarded as a reliable predictive biomarker for a pathological complete response and disease relapse in triple-negative breast cancer [22]. Our previous study also revealed that decreased VEGF after treatment is an independent prognostic factor for better PFS and OS in patients with esophageal squamous cell carcinoma receiving definitive CRT [21]. In the current study, better PFS and OS were also found in patients with HNSCC with decreased VEGF levels after TPF induction chemotherapy compared with those with increased VEGF values. This finding is compatible with previous studies on other cancer types.

In general, circulating VEGF levels are higher in patients with cancer than in healthy individuals, and elevated VEGF levels are associated with a poor response to treatment, disease progression, and poorer prognosis [37,38]. However, the ranges of VEGF values in patients with cancer and healthy individuals may overlap, resulting in the limitation of pretreatment VEGF values in terms of a prediction of response to tumor treatment. In addition, it is also difficult to define an optimal cut-off level of VEGF for clinical practice because the examination of VEGF in previous studies was performed using different methods at different institutions in different populations, contributing to the extensive variation observed in serum VEGF values. Therefore, post-treatment changes in VEGF abundance may represent a more useful tool to monitor responses to anti-cancer therapies in clinical practice, and this issue has been demonstrated in several studies [39,40,41]. In our study, the mean VEGF value in patients with HNSCC (150 pg/mL) was regarded as the cut-off level, according to a previous study [26]. Patients with post-TPF treatment VEGF levels ≥ 150 pg/mL were found to have poorer PFS and OS than those with VEGF < 150 pg/mL, and this finding was also compatible with the results of previous research.

The SDF-1α/CXCR4 axis has been reported to play an important part in the modulation of many responses, including chemotaxis, tumor growth, migration and distant metastasis [8,9]. Unlike VEGF, which is well investigated and frequently examined in several cancer types, SDF-1α is less applicated in clinical practice. However, SDF-1α may be valuable in the prediction of treatment response and prognosis. In general, carcinoembryonal antigen (CEA), the classic tumor marker, is higher in cancer patients compared to healthy humans. In a previous study, the median value of serum SDF-1α is 1501 pg/mL in esophageal cancer patients, and it is significantly higher than in healthy controls [25]. Therefore, the values of SDF-1α may be kinetic according to the treatment response to chemotherapy or radiotherapy, resulting in a difference in clinical outcome. In our study, we found SDF-1α decrease after TPF, and a lower post-TPF SDF-1α value (<1500 pg/mL) was related to better PFS and OS, suggesting SDF-1α is not only a diagnostic factor but also a prognostic marker.

In our study, we found that poor prognosis, including PFS and OS, were mentioned in patients with well-known poor prognostic factors, such as advanced T or N status, high tumor grade, negative HPV status, etc. In addition, patients without VEGF or SDF-1α decrease after TPF treatment, or higher post-TPF VEGF or SDF-1α values were also mentioned to have a worse prognosis, whether in the univariate or multivariate analysis. Moreover, we found that patients with both a VEGF and SDF-1α decrease after TPF treatment, and higher post-TPF VEGF and SDF-1α values had the worst PFS and OS than the other groups, indicating the role of the combination of serum VEGF and SDF-1α in the prediction of clinical outcome.

Our study had certain limitations. First, the patient sample size was relatively small, resulting in difficulties to present the statistical significance of this study, such as tumor location, clinical N status, and tumor stage. Second, selection bias caused by the relatively lower percentage of female patients (5.2%) may have existed. However, to the best of our knowledge, the present study is the first to explore the role of SDF-1α and VEGF in patients with HNSCC who received induction chemotherapy with TPF, and our findings may be helpful in predicting the prognosis of such patients in clinical practice.

## 5. Conclusions

Our study showed that SDF-1α and VEGF play crucial roles in HNSCC disease progression and that decreased SDF-1α and VEGF levels after TPF treatment, and lower post-TPF SDF-1α and VEGF values are independent prognostic factors for superior prognosis in patients with HNSCC who received induction chemotherapy with TPF followed by CRT.

## Figures and Tables

**Figure 1 biomedicines-10-00803-f001:**
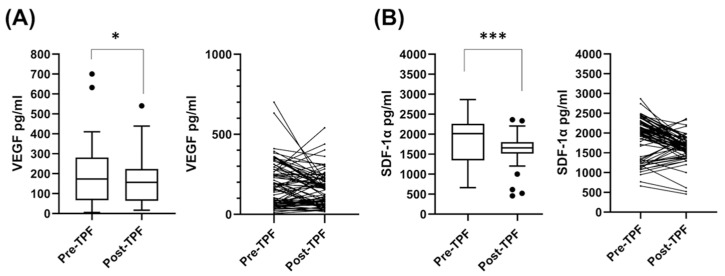
Comparison of serum SDF-1α and VEGF concentrations between pre-TPF and post-TPF treatments in head and neck squamous cell carcinoma patients. (**A**) The distribution and kinetic change of serum VEGF in the pre-TPF and post-TPF status. (**B**) The distribution and kinetic change of serum SDF-1α in the pre-TPF and post-TPF status. • means extreme values; * means *p <* 0.05 and *** means *p <* 0.001. SDF-1α: stromal cell-derived factor-1α; VEGF: vascular endothelial growth factor; TPF: docetaxel, cisplatin, and 5-fluorouracil.

**Figure 2 biomedicines-10-00803-f002:**
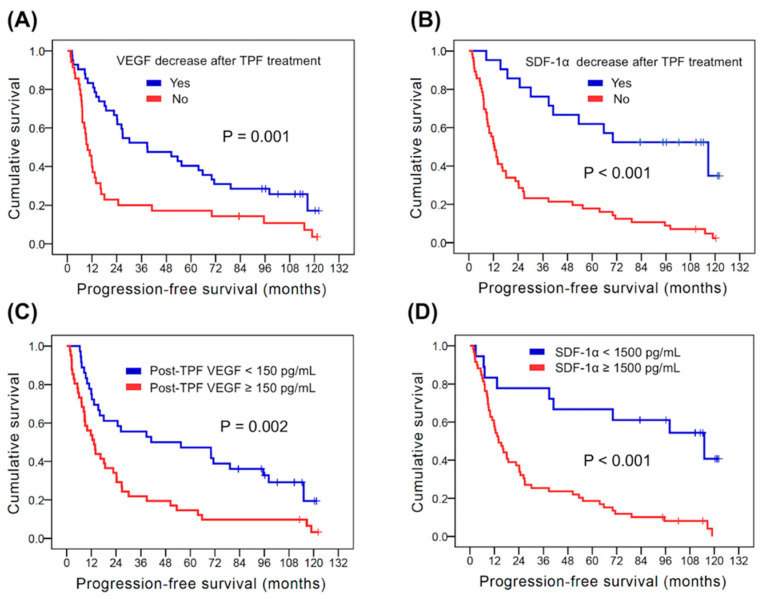
Kaplan–Meier curves of progression-free survival in head and neck squamous cell carcinoma patients. (**A**) The kinetic change of VEGF; (**B**) the kinetic change of SDF-1α; (**C**) the post-TPF VEGF values; (**D**) the post-TPF SDF-1α values. TPF: docetaxel, cisplatin and 5-fluorouracil; SDF-1α: stromal cell-derived factor-1α; VEGF: vascular endothelial growth factor.

**Figure 3 biomedicines-10-00803-f003:**
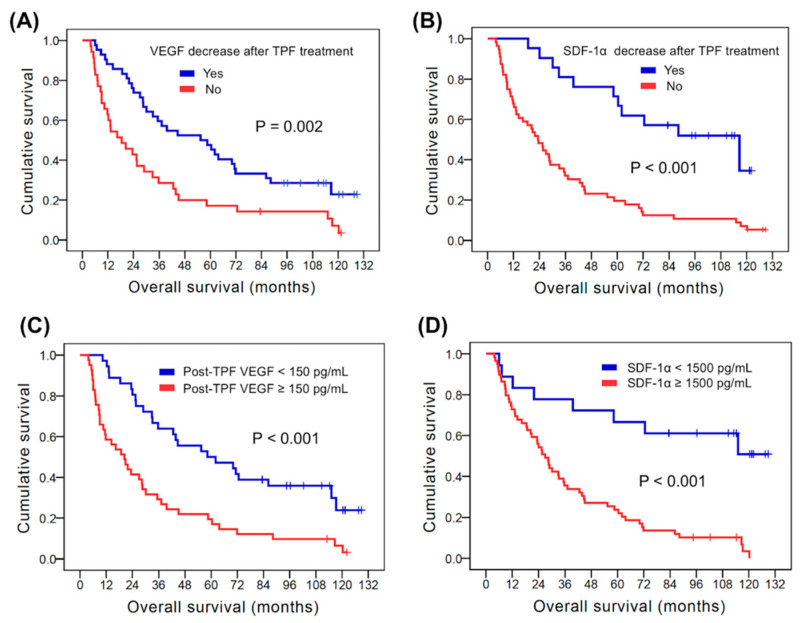
Comparison of Kaplan–Meier curves of overall survival in 77 patients with head and neck squamous cell carcinoma. (**A**) The kinetic change of VEGF; (**B**) the kinetic change of SDF-1α; (**C**) the post-TPF VEGF values; (**D**) the post-TPF SDF-1α values. TPF: docetaxel, cisplatin and 5-fluorouracil; SDF-1α: stromal cell-derived factor-1α; VEGF: vascular endothelial growth factor.

**Figure 4 biomedicines-10-00803-f004:**
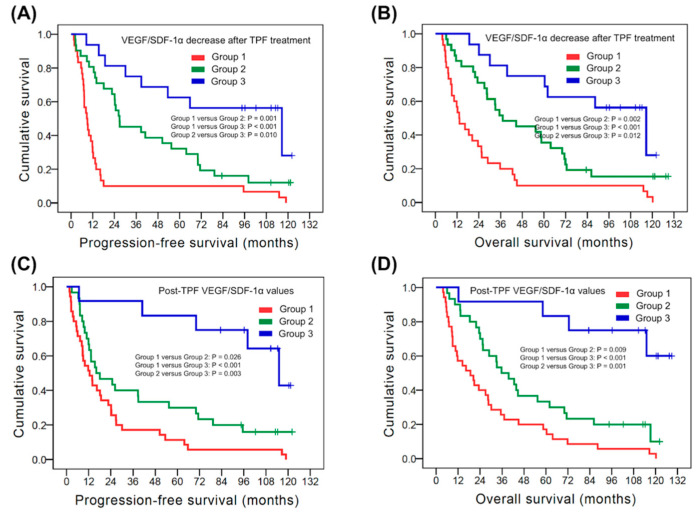
Kaplan–Meier curves of PFS and OS in patients with head and neck squamous cell carcinoma based on the combination of SDF-1α and VEG values. The kinetic change of SDF-1α and VEGF: PFS (**A**) and OS (**B**); the post-TPF SDF-1α and VEG values: PFS (**C**) and OS (**D**). PFS: progression-free survival; OS: overall survival; TPF: docetaxel, cisplatin and 5-fluorouracil; SDF-1α: stromal cell-derived factor-1α; VEGF: vascular endothelial growth factor.

**Table 1 biomedicines-10-00803-t001:** Characteristics of 77 locally advanced head and neck squamous cell carcinoma patients who received TPF induction chemotherapy followed by CRT.

Characteristics	
Age	53 years old (29–82)
Gender	
Male	73 (94.8%)
Female	4 (5.2%)
Location	
Oral cavity	22 (28.6%)
Oropharynx	28 (36.4%)
Hypopharynx	11 (14.3%)
Larynx	16 (20.7%)
HPV Status	
Positive	6 (7.8%)
Negative	71 (92.2%)
T Status	
2	19 (24.7%)
3	9 (11.7%)
4	49 (63.6%)
N Status	
0	18 (23.4%)
1	12 (15.6%)
2	41 (53.2%)
3	6 (7.8%)
Stage	
II	5 (6.5%)
III	7 (9.1%)
IVA	34 (44.2%)
IVB	31 (40.2%)
Grade	
1	19 (24.7%)
2	43 (55.8%)
3	15 (19.5%)

TPF: docetaxel, cisplatin and fluorouracil; CRT: chemoradiotherapy; HPV: human papillomavirus.

**Table 2 biomedicines-10-00803-t002:** Univariate and multivariate analysis of progression-free survival (PFS) in 77 patients with locally advanced head and neck squamous cell carcinoma patients who received induction chemotherapy of TPF followed by CRT.

Characteristics	No. of Patients	Univariate Analysis		Multivariate Analysis	
		PFS (Months)	*p*-Value	HR (95% CI)	*p*-Value
Age			0.23		
<60 years	65 (84.4%)	18.0			
≥60 years	12 (15.6%)	18.8			
Gender			0.41		
Male	73 (94.8%)	18.0			
Female	4 (5.2%)	22.6			
Location			0.06		
Oral cavity	22 (28.6%)	10.9			
Oropharynx + Hypopharynx + Larynx	55 (71.4%)	24.1			
HPV status			0.014 *		
Positive	6 (7.8%)	116.6			
Negative	71 (92.2%)	16.1			
T status			0.003 *		
2	19 (24.7%)	71.3			
3 + 4	58 (75.3%)	13.9			
N status			0.61		
0 + 1	30 (39.0%)	26.8			
2 + 3	47 (61.0%)	13.3			
Tumor stage			0.17		
II	5 (6.5%)	98.1			
III + IV	72 (93.5%)	16.4			
Grade			0.25		
1 + 2	62 (80.5%)	22.6			
3	15 (19.5%)	11.9			
VEGF decrease after TPF treatment			0.001 *		
Yes	42 (54.5%)	38.7		0.46 (0.27–0.52)	0.003 *
No	35 (45.5%)	9.9			
Post-TPF VEGF ≥ 150 pg/mL			0.002 *		
Yes	41 (53.2%)	13.3			
No	36 (46.8%)	41.0		0.50 (0.29–0.86)	0.011 *
SDF-1α decrease after TPF treatment			<0.001 *		
Yes	21 (27.3%)	116.6		0.38 (0.18–0.77)	0.007 *
No	56 (72.7%)	11.2			
Post-TPF SDF-1α ≥ 1500 pg/mL			<0.001 *		
Yes	59 (76.6%)	14.0			
No	18 (23.4%)	115.1		0.43 (0.19-0.95)	0.036 *

TPF: docetaxel, cisplatin and fluorouracil; CRT: chemoradiotherapy; HR: hazard ratio; CI: confidence interval; HPV: human papillomavirus; VEGF: vascular endothelial growth factor; SDF-1α: stromal cell-derived factor-1α. * Statistically significant.

**Table 3 biomedicines-10-00803-t003:** Univariate and multivariate analysis of overall survival (OS) in 77 patients with locally advanced head and neck squamous cell carcinoma patients who received induction chemotherapy of TPF followed by CRT.

Characteristics	No. of Patients	Univariate Analysis		Multivariate Analysis	
		OS (Months)	*p*-Value	HR (95% CI)	*p*-Value
Age			0.11		
<60 years	65 (84.4%)	35.3		0.47 (0.24–0.90)	0.024 *
≥60 years	12 (15.6%)	18.8			
Gender			0.40		
Male	73 (94.8%)	30.1			
Female	4 (5.2%)	43.8			
Location			0.032 *		
Oral cavity	22 (28.6%)	16.3			
Oropharynx + Hypopharynx + Larynx	55 (71.4%)	35.7			
HPV status			0.007 *		
Positive	6 (7.8%)	NR			
Negative	71 (92.2%)	28.6			
T status			0.003 *		
2	19 (24.7%)	71.3			
3 + 4	58 (75.3%)	25.6			
N status			0.26		
0 + 1	30 (39.0%)	44.7			
2 + 3	47 (61.0%)	21.8			
Tumor stage			0.05		
II	5 (6.5%)	NR			
III + IV	72 (93.5%)	28.6			
Grade			0.99		
1 + 2	62 (80.5%)	30.1			
3	15 (19.5%)	32.9			
VEGF decrease after TPF treatment			0.002 *		
Yes	42 (54.5%)	55.4		0.43 (0.25–0.74)	0.002 *
No	35 (45.5%)	18.4			
Post-TPF VEGF ≥ 150 pg/mL			<0.001 *		
Yes	41 (53.2%)	20.3			
No	36 (46.8%)	58.3		0.38 (0.22–0.65)	0.001 *
SDF-1α decrease after TPF treatment			<0.001 *		
Yes	21 (27.3%)	116.6		0.40 (0.20–0.83)	0.013 *
No	56 (72.7%)	23.3			
Post-TPF SDF-1α ≥ 1500 pg/mL			<0.001 *		
Yes	59 (76.6%)	27.0			
No	18 (23.4%)	NR		0.42 (0.18–0.95)	0.037 *

TPF: docetaxel, cisplatin and fluorouracil; CRT: chemoradiotherapy; NR: not reached; HR: hazard ratio; CI: confidence interval; HPV: human papillomavirus; VEGF: vascular endothelial growth factor; SDF-1α: stromal cell-derived factor-1α. * Statistically significant.

## Data Availability

The datasets used and analyzed during the current study are available from the corresponding author upon reasonable request.

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
