# Peer review of "Serum Levels of Stromal Cell-Derived Factor-1α and Vascular Endothelial Growth Factor Predict Clinical Outcomes in Head and Neck Squamous Cell Carcinoma Patients Receiving TPF Induction Chemotherapy"

_biomedicines, 2022, doi:10.3390/biomedicines10040803_

Round 1

Reviewer 1 Report

Generally a well written and interesting paper. A few comments:

Median PFS does seem short reflecting a poor prognosis group

Was HPV status collected?

Planning CT slices would include base of skull but perhaps this is what they mean by "upper neck". Not sure why you would scan down to upper pelvis.

Is it appropriate to use SDF-1a level from an esophageal trial?

Perhaps use their own medians for the analysis and also look at other cut-off levels. What counts as a fall after treatment?

Reviewer 2 Report

In this manuscript, the authors demonstrated that serum level of SDF-1α or/and VEGF are associated with prognosis in head and neck squamous cell carcinoma (HNSCC) patients who underwent TPF (docetaxel, cisplatin, and fluorouracil) induction chemotherapy. The advantages and clinical potential of using level of SDF-1α or/and VEGF after post-treatment as prognosis marker were well discussed. The limitations of this study were well discussed too. Please see below for some comments:

  1. What about the medical history of patients? Do they have similar medication histories or similar medication treatment besides the induction chemotherapy with TFP? Will those drug therapies affect the serum SDF-1α or VEGF level?
  2. What about the histological grading evaluation of HNSCC in these patients? Is there any correlation between histology grading with changes in SDF-1α or/and VEGF expression?
  3. Is there any correlation between SDF-1α and VEGF expression?

Reviewer 3 Report

The manuscript entitled “Serum Levels of Stromal Cell-derived Factor-1α and Vascular Endothelial Growth Factor Predict Clinical Outcomes in Head and Neck Squamous Cell Carcinoma Patients Receiving TPF Induction Chemotherapy” is presented in clear and well-structured way. The authors acknowledge the limitations of their study, in particular small patient sample size and relatively lower percentage of female patients. Despite the shortcomings, the work remains interesting, and the findings should be useful for clinical specialists and researchers. Still, some minor edition described below may improve the article.

Mistypes:

  • Lines 217, 218, 247, 248 – Should be “<” instead of “≥”.
  • Lines 303, 304 – No need to use italic.
  • Line 307 – Add space before brackets.

Suggestion:

Please describe the time points of sample collection. Protein biomarkers tend to have low clearance rate, so it is important to understand at what exact time frame after treatment serum can be collected.

Reviewer 4 Report

The manuscript is well-written and has some considerable interest to the researchers in the field. Some comments are below for authors' considerations

  1. As there is a separate section of materials please list all materials listed in this manuscript in one section
  2. Circulating levels of VEGF are induced by hypoxia and are angiogenic in vivo. Would the expression levels be influenced by ELISA measurements?
  3. Would it be possible to include western blot analysis for the VEGF for example?
  4.  As the sample population was 77 patients, why there were no healthy volunteers to compare the expression levels of VEGF and SDF1
  5. In the last paragraph of the introduction please expand on the use of TPF for non-MD readers
  6.  In table 1 would it be possible to incorporate the grading of the patients and the T stage, and the lymph node
  7. Figure 1 please enhance the axis labeling for readability and clarity 
  8. What was the patient enrollment approach adopted by the authors
  9. Were there any differences in the serum levels of VEGF before and after the SPF administration?
  10. Could the authors please comment on the survival and progressing response in the study samples

Round 2

Reviewer 2 Report

The authors have addressed all of my concerns.